# Short k-mer abundance profiles yield robust machine learning features and accurate classifiers for RNA viruses

Md. Nafis Ul Alam *, Umar Faruq Chowdhury

Department of Biochemistry and Molecular Biology, University of Dhaka, Dhaka, Bangladesh

* nafees3176@outlook.com

## Abstract

High-throughput sequencing technologies have greatly enabled the study of genomics, transcriptomics and metagenomics. Automated annotation and classification of the vast amounts of generated sequence data has become paramount for facilitating biological sciences. Genomes of viruses can be radically different from all life, both in terms of molecular structure and primary sequence. Alignment-based and profile-based searches are commonly employed for characterization of assembled viral contigs from high-throughput sequencing data. Recent attempts have highlighted the use of machine learning models for the task, but these models rely entirely on DNA genomes and owing to the intrinsic genomic complexity of viruses, RNA viruses have gone completely overlooked. Here, we present a novel short k-mer based sequence scoring method that generates robust sequence information for training machine learning classifiers. We trained 18 classifiers for the task of distinguishing viral RNA from human transcripts. We challenged our models with very stringent testing protocols across different species and evaluated performance against BLASTn, BLASTx and HMMER3 searches. For clean sequence data retrieved from curated databases, our models display near perfect accuracy, outperforming all similar attempts previously reported. On de novo assemblies of raw RNA-Seq data from cells subjected to Ebola virus, the area under the ROC curve varied from 0.6 to 0.86 depending on the software used for assembly. Our classifier was able to properly classify the majority of the false hits generated by BLAST and HMMER3 searches on the same data. The outstanding performance metrics of our model lays the groundwork for robust machine learning methods for the automated annotation of sequence data.

## Introduction

Viruses are numerous [1] and only a handful have been thoroughly characterized thus far [2]. As we have entered and slowly progress through the age of automated, high-throughput genomics, generation of sequence data itself is no longer of much concern, but rather accurate annotation of these sequences is the bottleneck in the expansion of biological knowledge. Remnants of viral genetics are continually integrated into host DNA where they reside as

**Data Availability Statement:** All data and links to resources is available at https://github.com/DeadlineWasYesterday/V-Classifier.

**Funding:** The author(s) received no specific funding for this work.

**Competing interests:** The authors have declared that no competing interests exist.

proviruses. Additionally, a copious number of viruses serendipitously accommodate our bodies as part of our microbiome (often called the virome [3]), deftly contaminate our working samples in the laboratory, and profoundly enrich the ecology of our environment. The scientific and pathological significance of viruses is astounding. The relevance of automated and reliable identification and characterization of highly divergent and novel viruses in laboratory samples and metagenomic data is now greater than ever.

The bacterial 16sRNA gene tremendously aids the study of bacterial phylogeny [4]. The lack of an omnipresent gene or genome segment that can be ubiquitously used to extract phylogenetically relevant data about viruses, complicates the study of virus evolution [5, 6]. There are additional levels to the structural complexity of viral genotypes. Viral genomes stand out from those of all other genomic entities as they can be either single, double or gapped DNA or RNA molecules with positive, negative or ambi-sense mechanism of genomic encoding, whereas all known realms of life exclusively rely on DNA as the genetic material. These categories of classification are built on top of the original Baltimore scheme [7] that sought to characterize viral genomes with regard to expression of genetic information [8, 9].

In the search for viruses in metagenomic data, researchers have noted that the usefulness of homology-based search tools are quickly exhausted [6, 10]. Most assembled contigs, likely to be from viral origins, are short and fragmented with no guarantee of containing coding regions that some of these tools rely on [11]. A number of software programs have been developed for the purpose of identifying viral sequences that have integrated into host genomes [12–15]. All machine learning models built for identification and characterization of viral sequences from cell samples or metagenomic data thus far rely on DNA sequences [16, 17]. RNA viruses comprise a major group having great clinical and scientific importance [18]. Presently made even more evident by the global pandemic caused by the 2019 novel SARS Coronavirus [19]. It has been demonstrated that RNA-Seq data can be a very promising avenue for improving knowledge on RNA viruses when leveraged by tactful algorithms [20]. Here, we present a novel short k-mer based scoring function that can be applied to sequences of different types to achieve impressive discriminatory capacity from classic machine learning models. With our scoring function, we were able to calculate the profiles of short genetic elements (k-mers) for RNA virus genomes and human RNA transcripts. We trained a number of classical machine learning classifiers on our sequence-derived numerical data in an attempt to classify assembled RNA sequences of unknown origin as either viral or cellular RNA. We tested the models rigorously using stringent cutoff parameters and many variable test sets including noisy de novo assembly data of cells cultured with Ebola virus. Furthermore, we demonstrated the flexibility of this approach by attempting to classify an RNA virus genome sequence from NCBI as either a positive-strand (ssRNA(+)) or a negative-strand (ssRNA(-)) RNA virus. We evaluated the performance of our selected model against nucleotide blast, protein blast and protein family HMMER3 search. In spite of the stiff testing protocols, our k-mer based numerical features stood out to be informative and robust datapoints for the classification of biological sequences by machine learning algorithms.

## Results

### Feature selection and scaling

A total of 5460 features were contrived using the sequence scoring formula for k-mer elements of length 1 to 6. The feature columns were tested for data normality by several methods. Q-Q plots of the data columns were suggestive of a normal distribution (not shown but can be found on git repository), but more robust statistical normality tests such as the Kolmogorov-Smirnov test and Shapiro-Wilk test strongly delineated a non-normal distribution for all

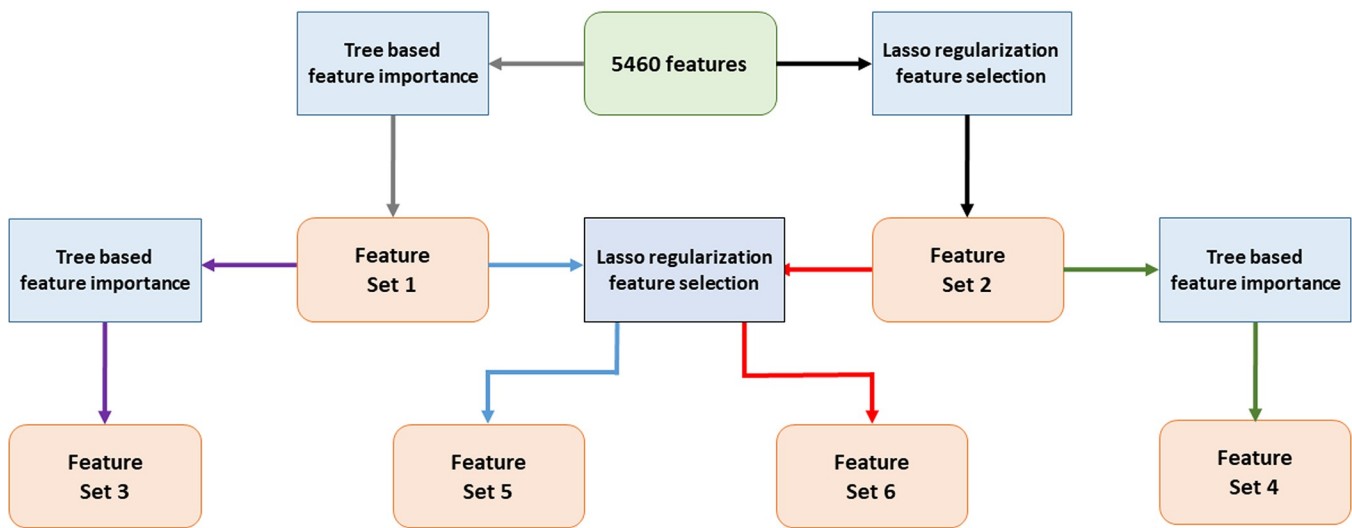

**Fig 1. Feature selection algorithms applied on succeeding levels to scale the model.** Tree-based and Lasso regularization-based feature selection applied on two levels successively yielded 6 total sets of features on the corresponding criteria of selection.

feature columns. Our features were therefore not suitable for univariate statistical analysis for feature selection.

The in-built, tree-based feature importance values of the scikit-learn library derived from our trained Random Forest classifier were applied for feature selection on two levels. Similarly, the scikit-learn optimized Lasso model applied to our trained Logistic Regression classifier with L1 penalty was also used for feature selection on two succeeding levels. Using a combination of both our tree-based selections and Lasso, we obtained a total of 6 levels of feature importance categories (4 from the trees and Lasso themselves and 2 more for each method leading into the other on a second level), each assigning a value of 1 to selected features and 0 to omitted ones (Fig 1). The top 3 feature sets ranked by importance from these 6 combined levels of selection are presented in Table 1. The complete table can be viewed in S1 Table. To simplify the model, make it more computationally scalable and remove noisy features, we selected 68 features that scored above 4 in our feature selection table. For comparison, models were trained on the 194 Features scoring above 3 as well, which includes the entirety of the features in Table 1.

## Performance of different classification algorithms

A total of six different types of classifier models were built- Logistic Regression Classifiers, Linear Support Vector Classifiers, Kernel based Support Vector Classifiers, Decision Tree Classifiers, Random Forest Classifiers and Gaussian Naïve Bayes Classifiers. For performance evaluation, all classifiers were trained thrice using 5460, 194 and 68 features accordingly. The train-test split was 25% and for cross-validation we applied 10-fold cross-validation on 9:1 fractions of the training set. The train-test split on each iteration was random. However, multiple iterations with new splits were carried out for all training events to ameliorate any random split biases.

**Table 1. Top 3 k-meric features sets when ranked by feature importance instated by our feature selection flow-chart.**

| Summed Feature Importance | Features |
|---|---|
| 6 | TTGACG, TAGCGT, ATACGC, CAAT, AAAAA, CCAAT, CATACG, TGAT, CGATA, AAAAAA, TCAATC, CAATTG, CGGTAA, GTTGA, GTTGAC, CGATAG, CCAATT, GGTTGA, ACCAAT, TCAA, TTGTCG, CGTCAA, CGGTTA, CAATCG, CGTTGA, TGTCGA, CGCAAT, CAAC, GTCGAA, GGTA, GTTG, GGTT |
| 5 | CAATC, TCCAAT, GCAATC, TTGAC, CGTAGT, ACGGTT, TTGA, GTTGGT, CATCAA, ATACGG, ATCGTA, CGATAA, CGATTA, GTTGAT, CGAT, CGATC, CGGTT, GCGTAC, CGCGTT, TTGCG, CGTCGA, CAATCT, ATCAAC, ATTGG, ATAGCG, GATC, TTGCGC, GGTTG, TCGGTT, GTCAAT, ATCAAT, TTCGAC, CCGTTA, CCGATA, TCGA, GCGTTA |
| 4 | ATCATA, ACAATC, CGTA, TCGT, ATGGTA, GGTAGT, TGGT, CTGT, GTCGA, AGCTG, GTCAA, AAATAA, TCCTG, CAGC, TATCCG, CTGGA, CAATT, TCGGTA, AGGAA, ATCAA, ATAACG, TCCAAC, AATTGG, AAATG, TGGTTC, AACC, TTGATG, GGCGTA, TGGTT, CGTTGC, GTCGAC, TCAAC, CAGA, TCAATT, CAGAG, TCAAT, AGGTTG, TCAACC, CGCGTA, ATCCAA, GGTACG, GTCATA, ATTGGT, GCATAC, GGTATG, GATTGG, GTACGC, AACCGA, TCGCAA, TTTTTA, ACATCG, GCGTAA, TTTTTT, ACCGGT, GTTGAG, TAACAC, ATAGGG, CGCAA, CCGCAA, CGGTTG, TATGGT, AACGGT, TAGGGT, CTAACG, CGGTA, GTTGT, GGTTC, ACGATA, TTCGCA, CTCGAT, TCGAGT, AAAATG, TCCAT, GTCAAA, AATAAA, TTGGCG, GTTGC, GGTCAA, CTGTA, ACGCAA |

The summed feature score is obtained by summing the total number of times the feature appears in the six filtered feature sets we obtained from our selection algorithm. The remainder of the table can be found in S1 Table.

For NCBI-viruses and human transcriptome data, all models displayed robust performance with high discriminatory capacity among the two classes (Table 2). The effects of feature reduction on the models can be viewed in the AUROC curves in Fig 2 and the S1 Fig. The poorest performing model was the Gaussian Naïve Bayes classifier trained on all 5460 features. The macro-averaged f1-score for the model was 88% and the binary f1-score was 81.83%. Among the others, the decision tree models with 5460 and 194 features and the Naïve Bayes model trained with 68 features were the only models to obtain binary f1-scores below 95%. All other models displayed resounding accuracy on both their cross-validation and test sets.

## Model selection

The best performing models were the logistic regression model with 194 features, the linear SVM model with 194 features and Kernel SVM models with 194 and 68 features, all displaying perfect classification accuracy on many random train-split-evaluate iterations (Table 2). The virus data set consisted of a total of 56 ssRNA(+) and ssRNA(-) families. It is common practice with such classifiers to carry out a separate cross-validation where the classifier is trained on the data set leaving out viruses belonging to a family to later use that family of viruses as an exclusive test set for evaluation [16, 21, 22]. We performed this leave-one-out cross-validation procedure for all 18 built classifiers. The complete results can be viewed visually on Fig 3 and numerically on S2 Table. Out of the four best-performing models previously mentioned, the logistic regression model with 194 features and the Kernel SVM model with 68 features displayed the best evaluation scores on the leave-one-family-out cross-validation test. Because the performance of the 194-feature logistic regression model is only slightly improved than the 68 feature Kernel SVM model, and because it is much more computationally demanding to

**Table 2. Performance metrics of 18 models fitted and the selected model.**

Part 1: Performance of 18 fitted models on cross-validation and test sets

| Model | Features | Correct/TP | False Positives | True Negatives | False Negatives | cv_bin[a] mean | cv_bin std[e] | cv_mac[b] mean | cv_mac std[e] | f1bin[c] | f1mac[d] |
|---|---|---|---|---|---|---|---|---|---|---|---|
| Logistic Regression | 5460 | 1049 | 4 | 3844 | 1 | 0.997559 | 0.001668 | 0.998457 | 0.001054 | 0.997622 | 0.998486 |
| Logistic Regression | 194 | 1020 | 0 | 3878 | 0 | 0.999194 | 0.001485 | 0.999489 | 0.000942 | 1 | 1 |
| Logistic Regression | 68 | 1025 | 10 | 3851 | 12 | 0.983759 | 0.005604 | 0.989727 | 0.003549 | 0.989382 | 0.993267 |
| Linear SVM | 5460 | 1010 | 2 | 3885 | 1 | 0.995825 | 0.000784 | 0.997351 | 0.000498 | 0.998517 | 0.999066 |
| Linear SVM | 194 | 1081 | 0 | 3817 | 0 | 0.99885 | 0.001281 | 0.999275 | 0.000808 | 1 | 1 |
| Linear SVM | 68 | 954 | 13 | 3922 | 9 | 0.985245 | 0.005137 | 0.990608 | 0.003267 | 0.988601 | 0.992902 |
| RBF Kernel SVM | 5460 | 995 | 1 | 3897 | 5 | 0.996293 | 0.00178 | 0.99765 | 0.001127 | 0.996994 | 0.998112 |
| RBF Kernel SVM | 194 | 1049 | 0 | 3848 | 1 | 0.99656 | 0.001552 | 0.997829 | 0.000978 | 0.999524 | 0.999697 |
| RBF Kernel SVM | 68 | 1012 | 0 | 3883 | 3 | 0.997572 | 0.001661 | 0.998463 | 0.001051 | 0.99852 | 0.999067 |
| Decision Tree | 5460 | 964 | 51 | 3817 | 66 | 0.939809 | 0.007824 | 0.961954 | 0.005977 | 0.942787 | 0.963846 |
| Decision Tree | 194 | 1006 | 67 | 3775 | 50 | 0.947326 | 0.006255 | 0.966403 | 0.004345 | 0.945045 | 0.964892 |
| Decision Tree | 68 | 1013 | 54 | 3786 | 45 | 0.943388 | 0.009383 | 0.964025 | 0.006727 | 0.953412 | 0.970253 |
| Random Forest | 5460 | 1032 | 0 | 3847 | 19 | 0.989437 | 0.003518 | 0.991985 | 0.002445 | 0.990879 | 0.994208 |
| Random Forest | 194 | 973 | 1 | 3911 | 13 | 0.991293 | 0.003575 | 0.994484 | 0.002069 | 0.992857 | 0.995535 |
| Random Forest | 68 | 987 | 3 | 3886 | 22 | 0.988948 | 0.003525 | 0.993217 | 0.001624 | 0.987494 | 0.992144 |
| Naïve Bayes | 5460 | 973 | 377 | 3493 | 55 | 0.81459 | 0.013953 | 0.877261 | 0.009596 | 0.818335 | 0.880049 |
| Naïve Bayes | 194 | 997 | 23 | 3867 | 11 | 0.982157 | 0.005434 | 0.988655 | 0.003468 | 0.983235 | 0.989429 |
| Naïve Bayes | 68 | 970 | 95 | 3812 | 21 | 0.942488 | 0.012554 | 0.963099 | 0.0081 | 0.94358 | 0.965126 |

Part 2: Performance of the Selected Model and our Random Forest Model on Divergent Data

| Model | Features | True Positives | False Positives | True Negatives | False Negatives | f1bin | f1mac | | | | |
|---|---|---|---|---|---|---|---|---|---|---|---|
| rbf-SVM68 | 68 | 1855 | 92 | 11259 | 677 | 0.82830989 | 0.897644 | | | | |
| RF68 | 68 | 1814 | 32 | 11319 | 718 | 0.8286889 | 0.898311 | | | | |

Part 1 lists the performance metrics on the test splits of the training data and Part 2 lists the performance on the divergent data comprised of viruses of other genotypes and mouse RNA transcripts.

[a] binary f1 score in cross-validation sets.

[b] macro averaged f1 score in cross-validation sets.

[c] binary f1 scores in test sets.

[d] macro f1 scores in test sets.

[e] standard deviation.

calculate 194 feature scores than to calculate 68, we select our 68 feature Kernel SVM model as the final model for ultimate evaluation on real data.

## Ability to generalize across species

The performance of the classifiers at this point far exceeded the outcomes of other classifiers built for the same task in previous studies [11, 16, 17, 21]. Given that the focus of these previous classifier were genomic DNA data, in order to test whether this performance amplification was entirely attributable to the greater homogeneity RNA sequences and their shorter lengths compared to genomic DNA sequences, we studied the performance of the classifier on

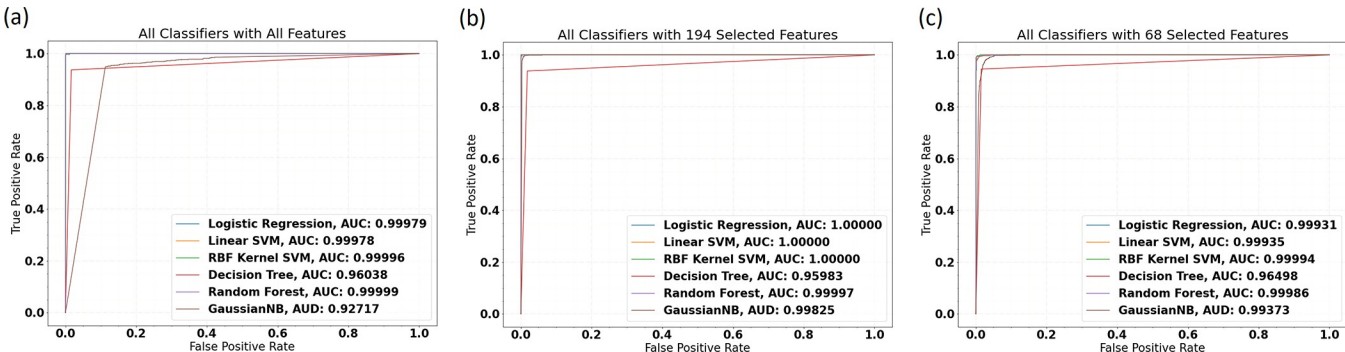

**Fig 2. AUROC for 18 fitted models trained while varying the number of features selected.** (a) Performance of all classifiers with all features. (b) Performance with 194 features after first round of selection. (c) Performance with 68 selected features. The weakest performing model was the Gaussian Naïve Bayes model displaying 0.96 AUC. No significant decline in model performance was noted when feature numbers were shrunk from 194 to 68.

systematically divergent data sets. We scored the sequences of total 77 other families (S3 Table) of viruses having DNA, cRNA and dsRNA genomes of similar length, and 11351

## Leave-One-Family-Out Cross-Validation Results

**Fig 3. Leave-one-family-out cross-validation results heatmap.** For every RNA virus family on the horizontal axis, there is a performance score for each model in the vertical axis. Other than the Decision Tree and Naïve Bayes models, performance was reasonably uniform all throughout the map.

putative-expressed RNA sequences of similar length from the *Mus musculus* whole transcriptome (S3 Table).

The 68 feature rbf-SVM model was able to accurately classify 13114 of 13883 test sequences with 1855 true positives, 92 false positives, 11259 true negatives and 677 false negatives with an AUROC of 0.95 (Fig 4). The f1-macro score was 0.89 and the f1-binary score was 0.83 (Table 2 Part 2). Performance of this magnitude on novel divergent data concerns one to be critical of the evaluation procedure and the presence of possible internal biases within the data set. However, this leaves out any possibility of overfitting of the model itself. Tree-ensemble models are known to stand out to be good discriminators in biological data [15, 23], particularly because of their propensity against overfitting. Our 68-feature random forest model displayed impressive classification accuracy and an AUROC of 0.96 on this divergent test set (Fig 4).

### Evaluation of performance against nucleotide BLAST

The unfaltering performance of the model in tests carried out thus far led us to suspect systematic biases present in the available RNA sequence data and however stringent, question our

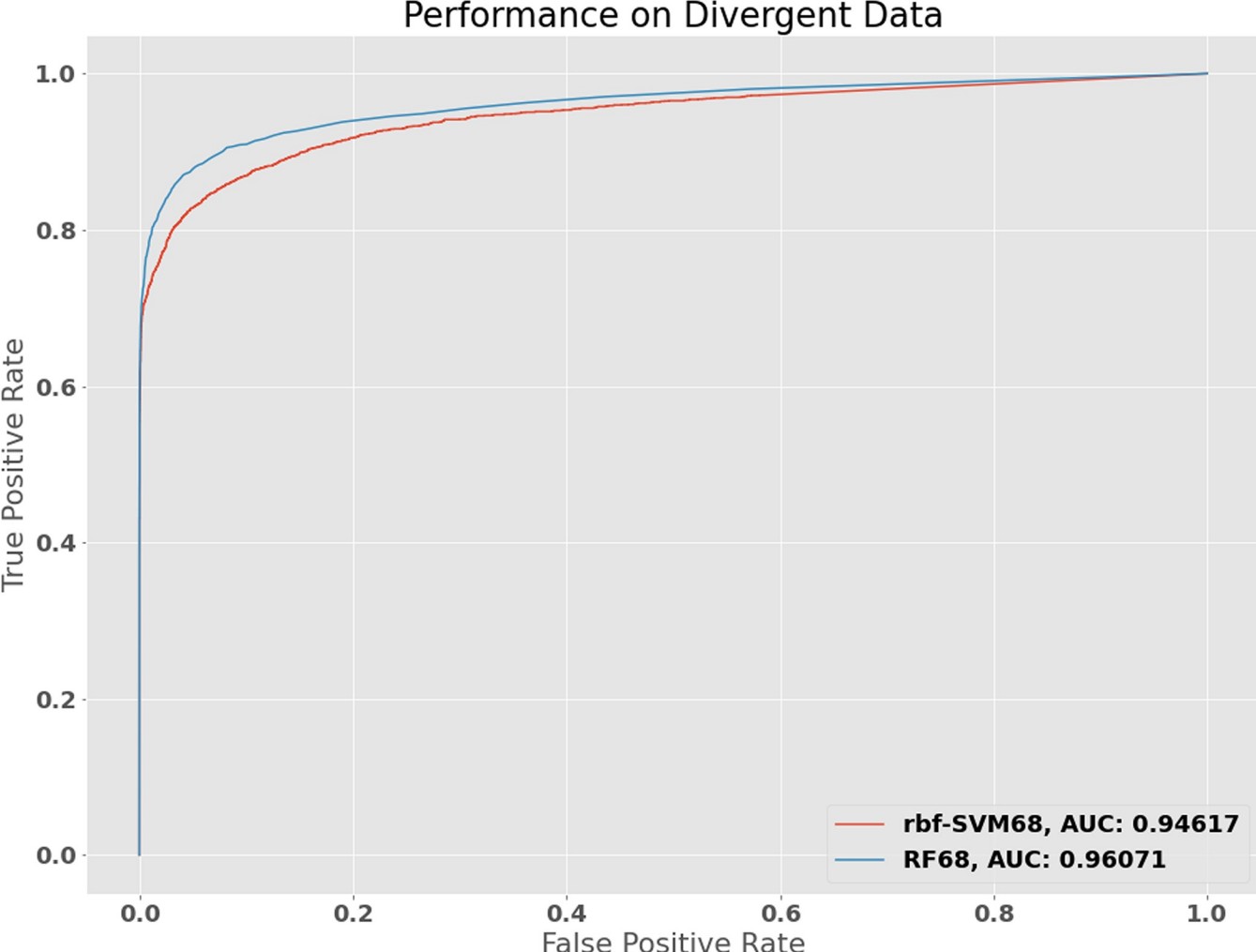

**Fig 4. Performance of the rbf-SVM68 and RF68 models on divergent data set.** To assess the capacity of our models to generalize to new examples, we attempted to confound the model with sequence data from viruses of other genotypes (having other than plus and minus RNA genomes) and RNA transcripts from mice (Listed in S3 Table). The robustness of the model performance metrics was still found to be intact.

testing protocols. For final evaluation it was indispensable that the model be put to the test against noisy real-word RNA-seq data lacking the meticulous curation that our previous data sets had. We collected de novo transcriptome assembly data for cultured human (*H. sapiens*), mice (*M. musculus*) and human cell lines incubated with an ssRNA(-) virus, Ebola virus from a previous cross-study on de novo assembly of RNA-seq data [24].

The best current standard for sequence-based taxonomic identification is the BLAST alignment algorithm [25]. Many correlated evolutionary forces govern both the conservation and divergence of sequence data, and as an artifact of these forces, even across the most divergent species, many gene families are shared and homologous sequences are found. This compromises the ability of sequence alignment protocols to properly label sequences that might have originated from highly divergent sources that are yet to be characterized and annotated in the current data repositories.

In the Homo sapiens transcriptome assembly, carried out on GM12878 human gamma-herpes virus 4 transformed B-lymphocyte cell lines, nucleotide blast query against our sequence database of ssRNA(+) and ssRNA(-) viruses yielded 135 false positives (S4 Table). These false positives were attributed to sequence regions of bovine viral diarrhea virus, giraffe pestivirus and a betacoronavirus that shared homology with some human ubiquitin gene transcripts and heat-shock protein 40 gene transcripts. Our 68 feature rbf kernel SVM model correctly classified 101 of these false positives giving us an accuracy of 75% on the unary data. When feeding the entirety of the human cell line assembly into our classification pipeline we achieved an accuracy of 63% for transcripts of all sizes and a much-improved value of 95% when taking transcripts of the same length range as our training data (Table 3). Similarly, nucleotide blast had given 130 false positives on the *Mus musculus* transcriptome assemblies (S5 Table), whereas our model accurate classified 93 examples with 72% accuracy on the unary class.

In the 23-hour Ebola virus with cells incubate, local nucleotide BLAST against our in-house RNA virus database had 10675 unique hits. Upon a second local nucleotide blast of these 10675 results against our human transcriptome training data, we obtained 3483 unique hits. Since we previously found only three virus sequences that originally had blastn hits against the whole human transcriptome, we labelled these second hits as belonging to the human class. This left us with a test set of 7146 positive class transcript examples for RNA viruses and 1172471 negative class examples of human transcripts in the whole 23-hour cell incubate with Ebola virus. The area under the ROC curve was 0.807 and 0.777 for all transcript sets and transcript sets of lengths resembling the training data respectively (Fig 5). The AUC improves more drastically when only human transcripts are filtered leaving all the virus transcripts still in the test set, owing to the minimized skew of the separate classes (Fig 5). The classification metrics from all instances are presented in Table 3.

Given the reported generalization prowess of the classifier from the original test set split and leave-one-out cross-validation, the classifier was expected to perform better on the real transcript data. Discordant with the previously observed success rate of the model, it performed better on the human samples compared to those of viral origin (Table 3). To further investigate the despondence of the model, we assessed the performance of the 10 assembly software separately. This revealed that performance varied significantly between the different assembly software that joined the contigs (Fig 6). Table 3 lists the performance metrics on the 10 assembly software based on virus/human labels instated from nucleotide BLAST results.

## Evaluation of performance against protein BLAST and HMMER3

For characterization of viral genomic data, protein BLAST or NCBI blastx is often used [26, 27]. Hidden Markov Model based multiple-alignment-profiling methods have also drawn the

**Table 3. Performance metrics of rbf-SVM68 model on real data.**

| data set | Transcript/ Software Filters | Total Number of Transcripts | True Positives | False Positives | True Negatives | False Negatives | f1binary | f1macro | AUROC | Unary Accuracy[a] |
|---|---|---|---|---|---|---|---|---|---|---|
| Unary Homo Sapiens Transcriptome Assembly | All | 6303331 | N/A | 2327082 | 3976249 | N/A | N/A | 0.38681 | N/A | 63% |
| Unary Homo Sapiens Transcriptome Assembly | Length Filtered | 146382 | N/A | 7067 | 139315 | N/A | N/A | 0.487632 | N/A | 95% |
| Homo Sapiens Cell Incubate with Ebola Virus Transcriptome Assembly | All | 1179617 | 5318 | 331849 | 540622 | 1828 | 0.03089 | 0.432644 | 0.799 | N/A |
| Homo Sapiens Cell Incubate with Ebola Virus Transcriptome Assembly | Both Class Length Filtered | 142803 | 49 | 3799 | 138906 | 49 | 0.024835 | 0.505587 | 0.776 | N/A |
| Homo Sapiens Cell Incubate with Ebola Virus Transcriptome Assembly | Only Human Length Filtered | 149851 | 5318 | 3799 | 138906 | 1828 | 0.684 | 0.817074 | 0.951 | N/A |
| Homo Sapiens Cell Incubate with Ebola Virus Transcriptome Assembly | bridger | 83744 | 16 | 20252 | 63452 | 24 | 0.001576 | 0.431906 | 0.6 | N/A |
| Homo Sapiens Cell Incubate with Ebola Virus Transcriptome Assembly | idba-tran | 78116 | 30 | 19860 | 58182 | 44 | 0.003005 | 0.42847 | 0.604 | N/A |
| Homo Sapiens Cell Incubate with Ebola Virus Transcriptome Assembly | soap-trans-default | 131711 | 19 | 44147 | 87538 | 7 | 0.0086 | 0.399727 | 0.793 | N/A |
| Homo Sapiens Cell Incubate with Ebola Virus Transcriptome Assembly | rna-spades | 206133 | 372 | 71543 | 134089 | 129 | 0.010274 | 0.39969 | 0.775 | N/A |
| Homo Sapiens Cell Incubate with Ebola Virus Transcriptome Assembly | spades | 93814 | 4488 | 25645 | 62559 | 1122 | 0.251126 | 0.537447 | 0.843 | N/A |
| Homo Sapiens Cell Incubate with Ebola Virus Transcriptome Assembly | trinity | 97849 | 32 | 22936 | 74839 | 42 | 0.002778 | 0.434846 | 0.611 | N/A |
| Homo Sapiens Cell Incubate with Ebola Virus Transcriptome Assembly | oases | 178498 | 180 | 34721 | 143373 | 224 | 0.010197 | 0.450784 | 0.691 | N/A |
| Homo Sapiens Cell Incubate with Ebola Virus Transcriptome Assembly | trans-abyss | 213005 | 171 | 68268 | 144346 | 220 | 0.004969 | 0.406611 | 0.583 | N/A |
| Homo Sapiens Cell Incubate with Ebola Virus Transcriptome Assembly | binpacker | 12291 | 7 | 800 | 11468 | 16 | 0.016867 | 0.491256 | 0.657 | N/A |
| Homo Sapiens Cell Incubate with Ebola Virus Transcriptome Assembly | shannon | 84456 | 3 | 23677 | 60776 | 0 | 0.00253 | 0.418611 | 0.862 | N/A |

[a] Unary accuracy is used to assess performance for test sets containing only one class of examples and thus rendering f1 scores uninformative.

interest of researchers for the task [22, 27, 28]. These methods rely on primary protein sequences for distinction between virus and non-virus data. Although protein primary sequences are more reliable and informative because they lack noisy repeats and 'junk' that is characteristic of nucleotide sequence data, the dichotomy for their use in taxonomic classification lies in the fact that they share more homology across species than general sequence data. For these protein-based detection methods, it is common procedure to first weed out the

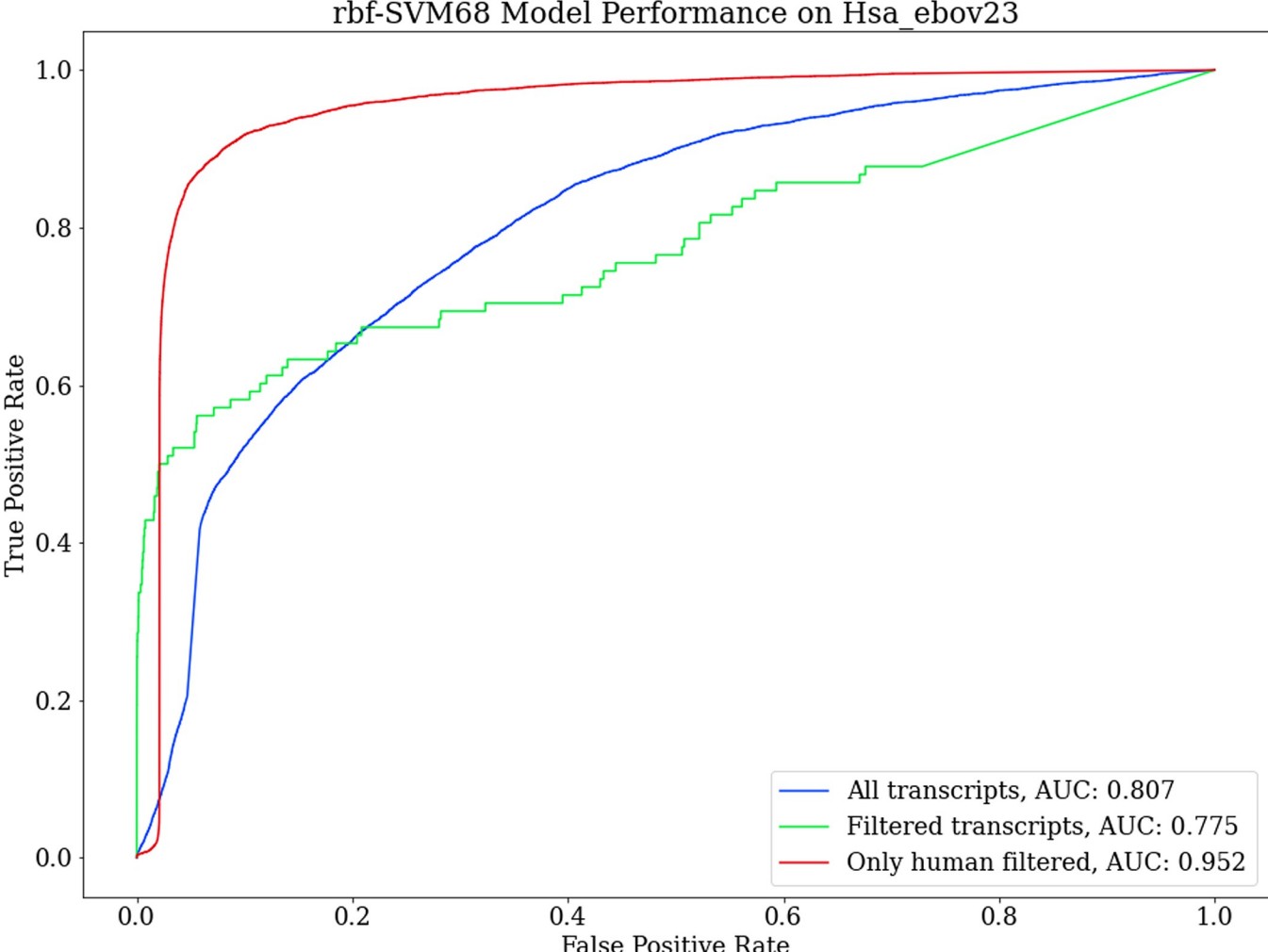

**Fig 5. Performance on RNA-Seq assembly data from human cells cultured with Ebola virus.** The model performed the best with AUC of 0.95 when human transcripts below the training range length were left out. Filtering the short virus transcripts out as well as the human transcripts resulted in a drop in the curve owing to the skew in the number of examples belonging to the human class.

profiles/protein-families against a negative control as they would otherwise display a great propensity towards false positives hits [20].

BLASTx employs a built-in 6 frame-translation algorithm. In order to gain more control over the protocol, we built a Python based in-house six-frame ORF translator with biopython that uses the standard genetic code table (ncbi-translation table ID 1). Short ORFs that are not part of any genes, do not code for proteins and occur entirely due to chance are dispersed throughout genomes. By plotting the frequency of proteins discovered against the protein length cutoff threshold, it was evident that at a length of about 100 amino acids, non-genic random ORF numbers significantly subside (Fig 7). Using this 100-amino-acid cutoff for 6-frame in-silico translation, we translated our entire ssRNA sequence data set and all human de novo transcriptome assemblies to proteins. Querying the complete human proteome for these translated protein sets yielded vast numbers of false positives with very low expect values. We found 1083 in-silico translated viral proteins to have hits with human proteins with a 0.01 e-value cutoff and 27023 proteins with no cutoff. We sifted out these 1083 proteins showing multiple

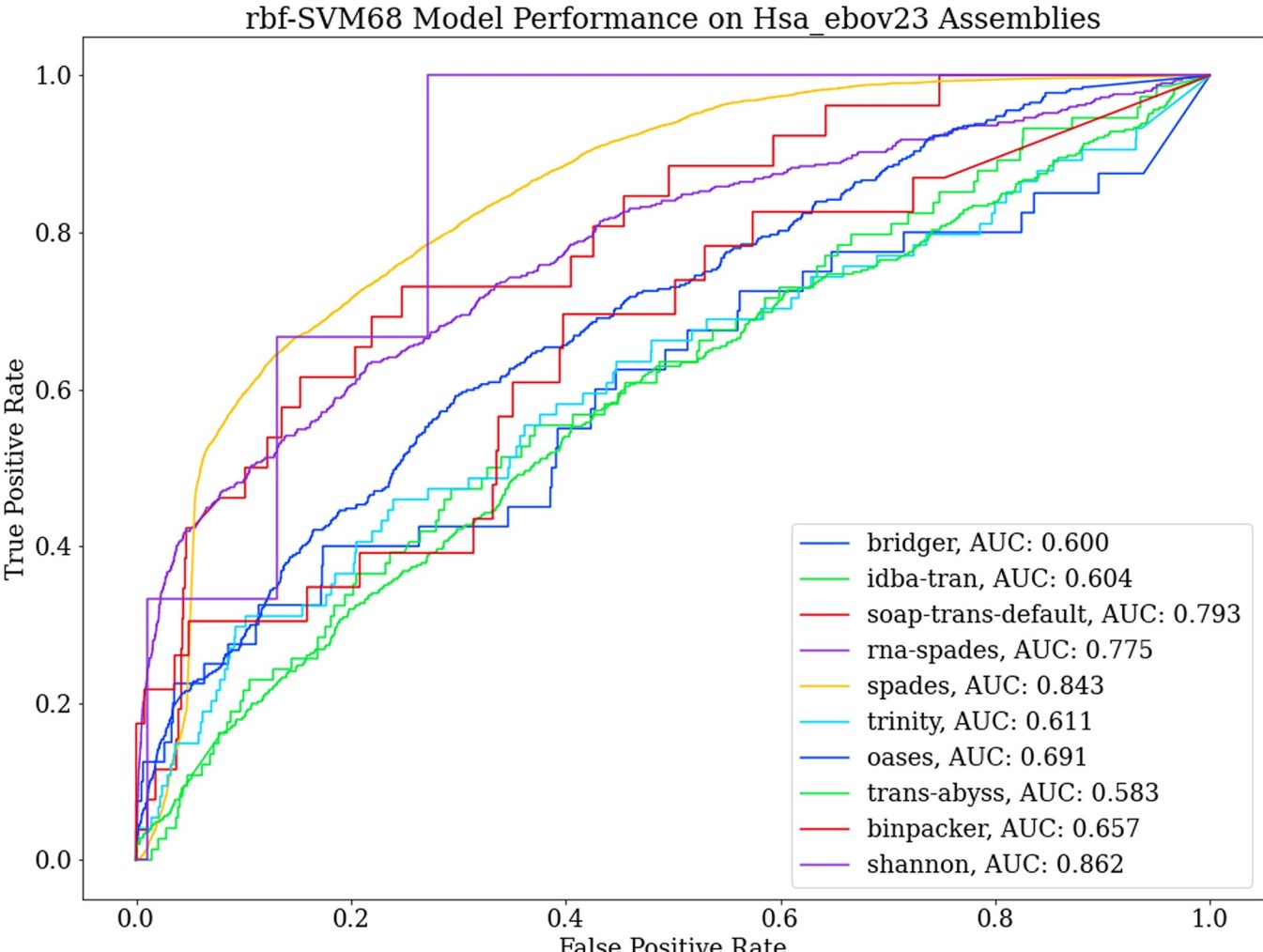

**Fig 6. Performance on RNA-Seq assembly of human cells cultured with Ebola virus based on the software for assembly.** There is significant variation in performance across the different assembly software. Spades had the most uniform performance where performance was modest whereas more popular assembly tools such as trinity, oases and trans-abyss had poorer but similarly uniform performance.

uninformative hits and created a new in-silico viral protein database of only informative proteins. We were left with 38483 predicted proteins in the database. blastp searches on our informative in-silico viral protein database yielded 505794, 3392 and 34 false positives with cutoffs of 10, 0.01 and 0.0001 respectively. Our rbf-SVM68 classifier showed a modest 62%, 64.2% and 70.6% recall in classifying these false hits as human transcripts in the unary data set (Table 4).

In the same manner, we sorted out 2346 uninformative vFams from the high performance vFams vFam A available at http://derisilab.ucsf.edu/software/vFam/. Using hmmsearch on our informative vFams against the de novo assembly data set yielded 26003 false hits. Our model was able to correctly classify 14089 of these false hits as human transcripts as noted in Table 4.

## Binary classification of plus and minus-strand RNA viruses

To demonstrate the flexibility of our computational pipeline for sequence classification, we separately evaluated the selected model's ability to distinguish positive and negative sense RNA viruses with 5460, 194 and 68 features respectively. The models were able to very

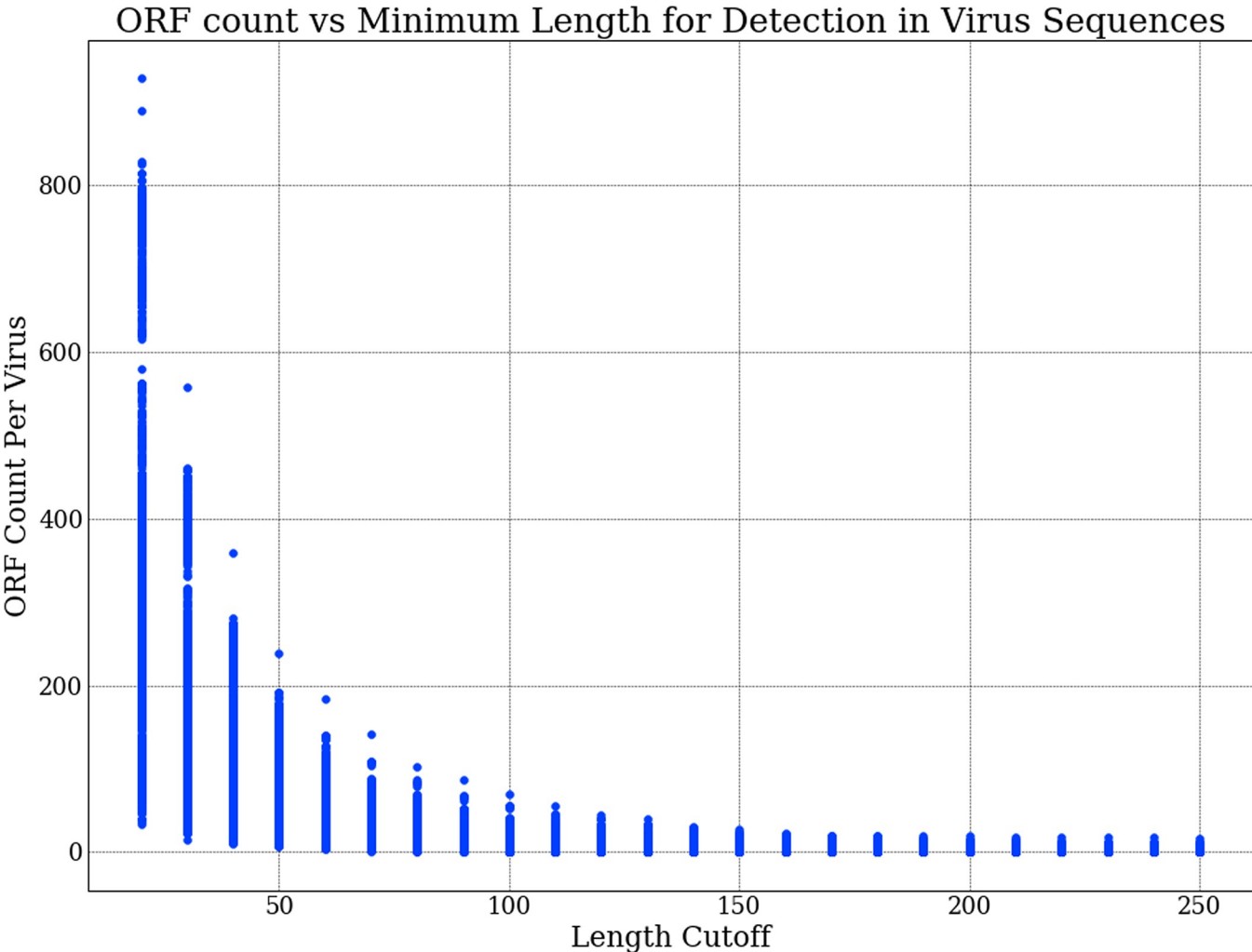

**Fig 7. Visualizing count of detected ORFs in RNA virus sequences changing in relation to the minimum length cutoff that was set.** We would expect all viral genomes to encode a minimal set of proteins necessary for successful replication and the number of these proteins to deviate minimally across viruses of different types. The graph depicts that at a cutoff range between 100 and 150, we achieve the optimal cutoff point where only important ORFs that have greater probability of being involved in the information pathway of the virus are considered.

**Table 4. Performance evaluation of rbf-SVM68 against BLAST and HMMER3.**

| Tool | Cutoff | Query | Subject | False Hits | Our Model | False Positives | True Negatives | Accuracy |
|------|--------|-------|---------|-----------|-----------|-----------------|----------------|----------|
| BLASTn | None | Human Transcriptome De novo Assembly | RNA Virus Sequence Database | 135 | rbf-SMV68 | 34 | 101 | 75% |
| BLASTn | None | Mouse Transcriptome De novo Assembly | RNA Virus Sequence Database | 130 | rbf-SMV68 | 37 | 93 | 72% |
| BLASTx | None | Human Transcriptome De novo Assembly | Informative Proteins Only | 505794 | rbf-SMV68 | 191958 | 315528 | 62% |
| BLASTx | 0.01 | Human Transcriptome De novo Assembly | Informative Proteins Only | 3392 | rbf-SMV68 | 1214 | 2178 | 64.20% |
| BLASTx | 0.0001 | Human Transcriptome De novo Assembly | Informative Proteins Only | 34 | rbf-SMV68 | 10 | 24 | 70.60% |
| hmmseach on informative vFams | None | Human Transcriptome De novo Assembly | Informative vFam Profiles Only | 26003 | rbf-SMV68 | 11914 | 14089 | 54% |

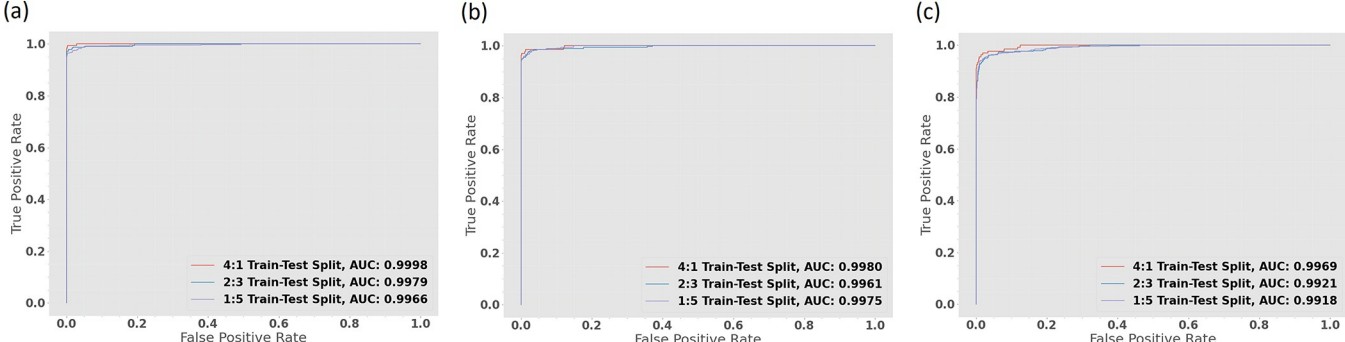

**Fig 8. Performance of the rbf-SVM model trained with the same computational pipeline for classifying positive and negative sense RNA viruses.** (a) Performance on different train-test splits when trained with all 5460 features. (b) Performance when trained with 194 features. (c) Performance when trained with 68 selected features. All models exhibit a great ability to specify the different sequence classes demonstrating the flexibility of our pipeline.

precisely classify positive and negative sense RNA viruses even when trained on a minimalist data set with a train-test divide of 1:5 (Fig 8). The macro averaged f1scores were above .95 in all test cases and the macro scores for the cross-validation set were also convincingly high with low standard deviation. The classification reports are delineated in Table 5. Fig 9 delineates the complete computational pipeline utilized in this study.

## Discussion

The concept behind this study originated from similar k-mer based prototypes we built for classifying positive and negative sense RNA viruses. We later found that in the past, Random Forest classifiers for 16S rRNA classification based on similar k-mer based principles had displayed outstanding performance metrics [29]. Machine learning attempts at sequence classification have been sparse in the past. But as of late, there is more activity in the field primarily due to advances brought about by next-gen sequencing and growing interest in metagenomics [21, 30, 31], symbionts [11] and microbial colonies [28]. Bzhalava et al. [21] attempted to detect viral sequences in human metagenomic data sets and although their classifier found very promising contrast on GenBank data, the model performed very poorly on a real metagenomic data set. Our models trained on NCBI data displayed 0.8 to 0.95 AUROC on real test data (Fig 5), which is a big improvement form their AUROC of 0.51. k-mer based Random

**Table 5. Performance of rbf-SVM in classifying positive and negative sense RNA viruses.**

| Number of Features | Train-Test Split | True Positives | False Positives | True Negatives | False Negatives | cv_binary mean | cv_binary std | cv_macro mean | cv_macro std | f1binary | f1macro |
|---|---|---|---|---|---|---|---|---|---|---|---|
| 5460 | 4:1 | 145 | 0 | 1422 | 6 | 0.972715 | 0.012229 | 0.985083 | 0.006681 | 0.97973 | 0.988812 |
| 5460 | 2:3 | 308 | 0 | 3445 | 21 | 0.966849 | 0.028873 | 0.981788 | 0.015834 | 0.967033 | 0.981997 |
| 5460 | 1:5 | 416 | 0 | 4587 | 29 | 0.950275 | 0.056018 | 0.97274 | 0.03045 | 0.966318 | 0.981584 |
| 194 | 4:1 | 125 | 0 | 1435 | 13 | 0.9759 | 0.013335 | 0.986788 | 0.007303 | 0.95057 | 0.973031 |
| 194 | 2:3 | 298 | 0 | 3453 | 23 | 0.968434 | 0.010478 | 0.982572 | 0.005786 | 0.962843 | 0.979762 |
| 194 | 1:5 | 389 | 0 | 4575 | 68 | 0.924052 | 0.043204 | 0.95789 | 0.023325 | 0.919622 | 0.956122 |
| 68 | 4:1 | 125 | 3 | 1430 | 15 | 0.945588 | 0.026894 | 0.970244 | 0.014663 | 0.932836 | 0.963291 |
| 68 | 2:3 | 281 | 6 | 3444 | 43 | 0.924708 | 0.031719 | 0.958641 | 0.01733 | 0.919804 | 0.95637 |
| 68 | 1:5 | 383 | 9 | 4575 | 65 | 0.896827 | 0.065865 | 0.943638 | 0.035787 | 0.911905 | 0.951941 |

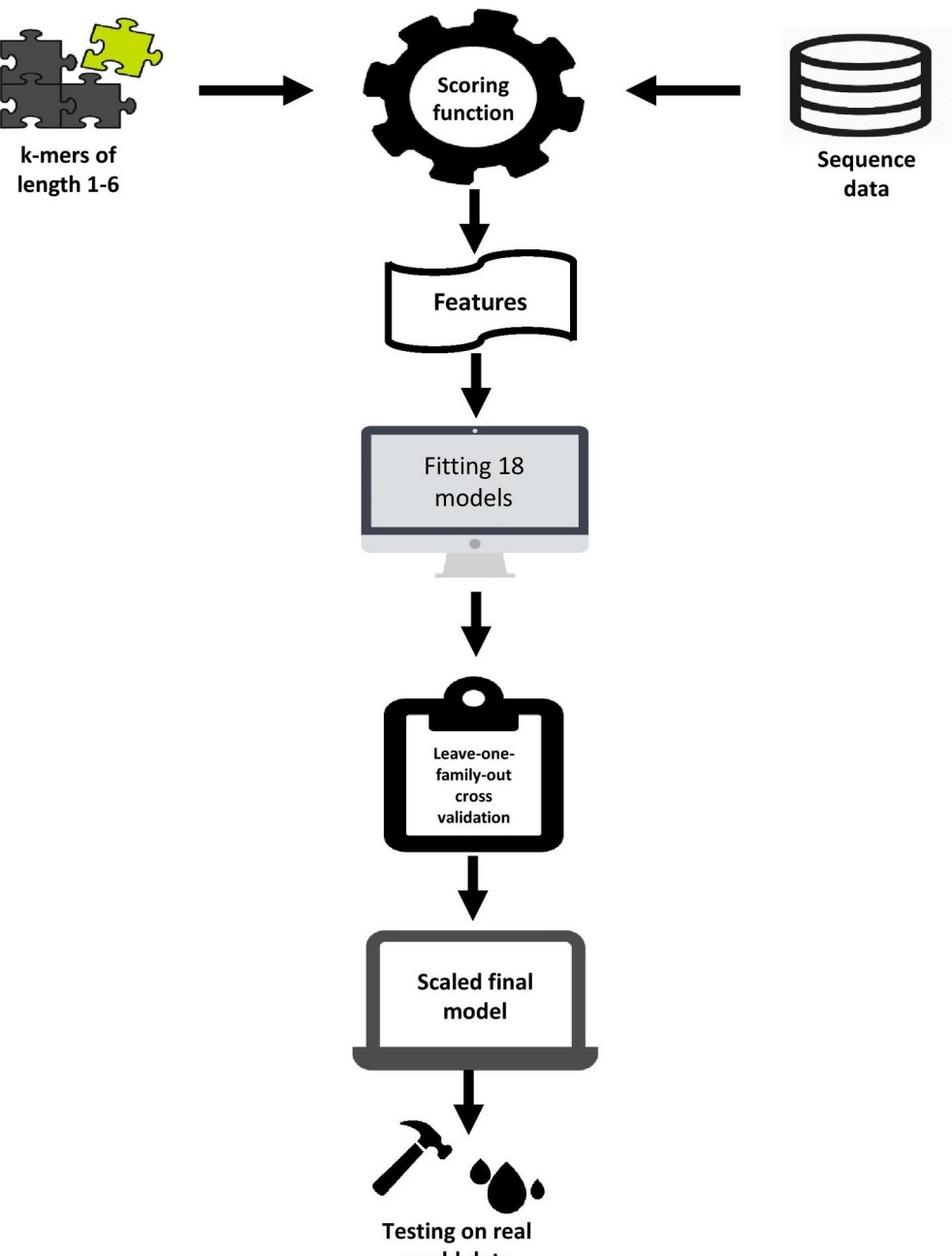

**Fig 9. Diagrammatic representation of our complete experimental design.**

Forest models by the same team using k-mers of lengths 1 to 7 achieved the best AUROC of 0.875 with k-mers of length 6 [16], which is significantly lower than the metrics of our Random Forest models. VirFinder is a k-mer based machine learning tool that can identify prokaryotic viral contigs from metagenomic data [11]. The performance of VirFinder was evaluated with varying k-mer lengths, class skews and contig lengths. Our models trained on varying numbers of features and large class discrepancies display more consistent and robust performance metrics on average across all tests concerning NCBI data. The AUROC for our models on the

divergent data set comprised of mouse transcripts and DNA viruses was 0.95 to 0.96 compared to their 0.90 to 0.98 on an analogous simulated data set.

The features we constructed from short k-mers displayed strong discriminatory capacity among the sequence classes. This has stood out to be the case in all tests carried out in this study including those that were left out from the final paper. Both positive and negative sense RNA viruses must produce complementary strands of reverse polarity in order to replicate. In positive sense viruses, the reverse complements are necessary for making new copies of their genome since all polymerases are unidirectional from the 5' end to the 3' end and in negative sense viruses the necessity of making reverse complements goes beyond DNA replication as their primary genetic information pathway relies on the synthesis of the reverse complement that codes for viral proteins. It felt necessary to include the reverse complements of the viral genomes as we would undoubtedly encounter remnants of them in whatever virus infected cell we sought to explore. The increase in training examples did not hurt the model by relegating bias upon it, as evident from the generalization tests.

The scikit-learn library offers feature importance scores from the Random Forest tree ensemble. It is calculated from the selection criterion based on the gain of information by the split at the nodes. We selected the entropy criterion as our measure of information gain because of the heterogeneous nature of our features. The disadvantage of tree-based feature selection is the decrease in weight assigned to correlated features. The smaller k-mers are all slices from the larger k-meric words and hence the are highly correlated. We thus sensed the need for going beyond solely tree-based selection approaches. The only drawback of our model stood out to be the moderate computational power necessary to calculate all the features. For short RNA virus sequences, it took us a few minutes on a 4 core third-generation i5 computer. Based on our results, the design of the model proved to be highly flexible and we make a strong case for the use of similar k-mer scoring profiles for classification of other classes of sequence data. It was thus necessary to scale down the model by reducing the number of features so it could comfortably adapt to much longer sequence classes. In order to scale the model, we went beyond tree-based methods and employed Lasso regularization. Feature selection by Lasso stems from linear regression models by penalizing the coefficients of the variables. Using a combination of these selection methods on multiple levels, we were able to shrink the model down into only few features while only minimally compromising performance.

Overfitting is always a concern when applying ML models to biological data. The first impressions on the high accuracy of the classifiers suggested a case of overfitting in the model. The bootstrap aggregating or bagging meta-algorithm employed in Random Forest classification is particularly resistant to high variance and overfitting. Although Decision Trees employ the same bagging principle, because of their larger depths, they are more prone to higher variance and overfitting. Many random sampling events generate subsets of data known as bootstrap samples. Isolated Decision Trees are trained on these bootstrap samples to construct the Random Forest ensemble. A majority vote from the members of the ensemble generates predicted classifications for test sets. Since our random forest models performed noticeably better than the decision tree model, it would lead us to believe that overfitting does not account for the fidelity of our models. Stronger evidence against a case for high variance comes from the model's impeccable performance when tested against viruses not belonging to the RNA virus groups and mRNA from mouse transcriptomes. These test sets substantially varied from the training data and a highly biased model would have displayed a moderate decrease in performance. High performance metrics on very small training sets in our positive and negative sense RNA classification tests are also good evidence against high bias in the model.

Our Gaussian Naïve Bayes models had the poorest performance metrics. This is easily explained by heterogeneous and nonuniform nature of our features. When tested for gaussian normalcy, all features failed to show a distribution resembling the normal distribution. Hence, the performance of the Gaussian Naïve Bayes models was poor owing to the non-continuous datapoints in our feature sets. The improvement in its performance after the reduction in the number of features is suggestive of a non-linear nature of the numerical values and the presence of random noise, which is customary to biological data sets. The superior performance of Logistic Regression models and both Support Vector Machine models suggest a discontinuous polar segregation for the feature values in their respective classes.

When progressing to real RNA-Seq assembly data, the previously robust, infallible performance of the model began to wane. In the assemblies from the human cell line, filtering the transcripts to conform to lengths resembling the original training set solved the issue and the robustness of the classifier became apparent again. But yet, performance on the 23-hour Ebola virus cell incubate was less impressive. This can be easily explained by the incongruity of the assembly software, the unwanted mixing of contigs from either source during assembly and the irregularities in the lengths of the assembled contigs. The false negative results were also higher than we expected. It would have been a matter of concern if these misclassifications were a result of the very high mutation rates characteristic of RNA viruses [32], but the evidence from the cross-validation and generalization studies and error analysis within the real data set (not discussed) lead us to believe that such is not the case. We postulate that as de novo assembly tools improve, the classification performances of our models would tend towards that observed in curated sequence data. Since all human training data was composed of gene transcripts, whereas the sequences in the virus training set were mostly comprised of non-genic regions, we hypothesized that this could be a cause for greater numbers of false negatives because the majority of virus sequences in the cell-virus incubate would also inevitably be viral gene transcripts as a consequence of the infectious activity of the Ebola virus, rather than the whole genome or sequencing with long non-genic regions. However, including RNA virus transcripts separately downloaded from NCBI into the training set only marginally improved performance on the Ebola virus and cell incubate sample (not discussed). This was more evidence to support our claim that the poor performance is greatly justified by shortcomings in the assembly of the RNA-Seq data itself.

## Conclusion

In this study, the use of short k-mer profiles as machine learning features proved to be very resourceful for training sequence classification models. Machine learning and deep learning models that train on sequence data more commonly treat biological sequences as lifeless vectors of binary information. The sequence of nucleotide bases in nucleic acids does not occur randomly and independently with identical probability. Rather, a confluence of biophysical and biochemical forces actively dictates the nature of this evolution. Our results question this approach for training sequence classifiers and sheds light on the elegance in the emergence and evolution of genomes centered around short k-meric repeats. The evaluation metrics from the classification results in the leave-one-family-out experiments, the performance on the divergent sequences and the ability to differentiate plus and minus strand viruses even when trained on very small data sets, is good evidence that is greatly suggestive of a robust set of numerical machine learning features. The pipeline depicted on Fig 9 can be applied to any two classes of sequence data having considerable taxonomic distance to train machine learning models that can accurately report the presence of either class in unknown sequence data.

# Suggested Approach for Acquiring Best Automated Annotations

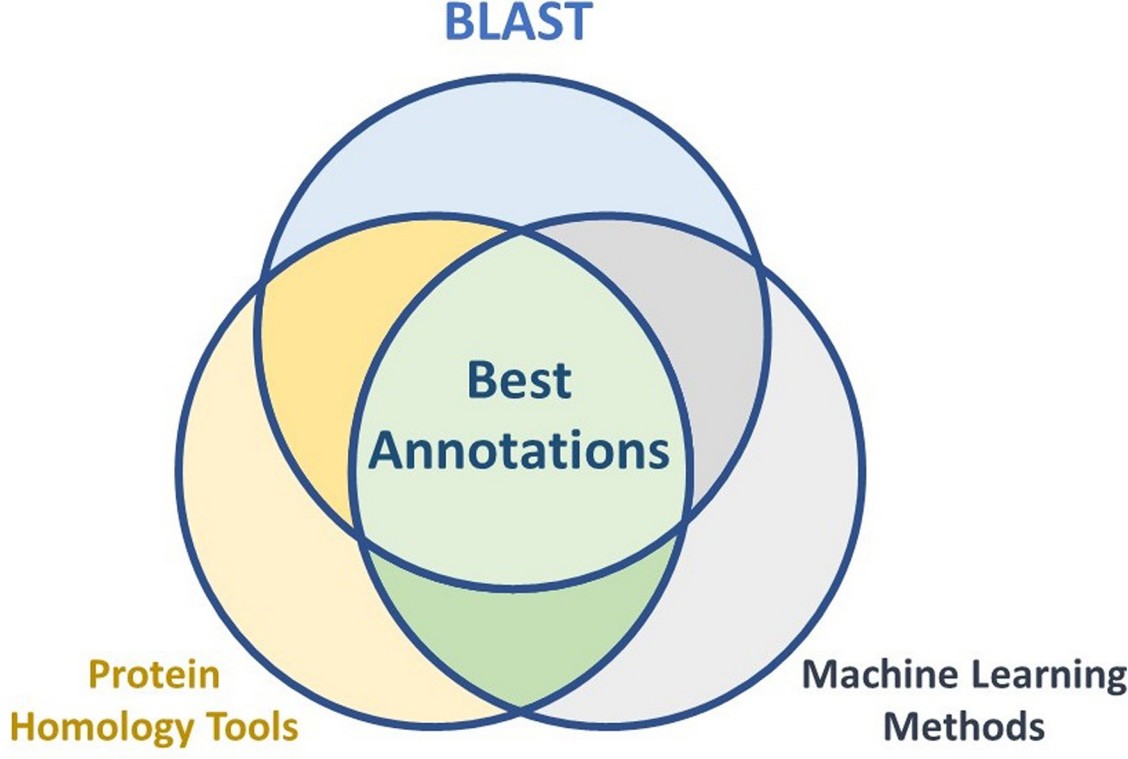

**Fig 10. Suggested approach for acquiring the best automated annotations.** Our explorations indicate that there is no single annotation tool that can produce the best results without requiring further curation. Perhaps the best way to use these tools is by stacking the results on top of each other to remove maximum number of false positives while consolidating the true positive results.

Nucleotide blast is still the best tool at our disposal for characterization of unknown sequence data. But due to the homology omnipresent in biological sequences even between distinct orders of life, false positives and ambiguous results are always prevalent. The drawbacks of this approach include, the very limited number of viral genomic diversity that has been so far catalogued, the uncertainty of there being a known homolog or close ancestor in the database, and the prevalence of many viral genomic remnants in the genomes of host organisms. We can also reasonably conclude that protein-based methods such as protein blast or pfams are not suited to the task of taxonomic classification of greatly divergent RNA sequences. Pertaining to the nonuniformity of the overlap between the results from our classifier and the tools it was compared to, we propose that a combinatorial approach employing multiple tools simultaneously is currently the best option for segregating sequence data by taxonomy (Fig 10).

De novo assembly of RNA-Seq data facilitates the discovery of novel transcripts and splice variants. It can offer great promise in the discovery of novel RNA viruses without prior sequence information. Software packages that carry out de novo assembly of RNA-Seq data are still improving. The 10 assembly tools we tested our model on were still highly variable in spite of assembling the same raw sequence data. For the identification of divergent viruses, the use of robust and highly specific methods such as PCR, immunoassays or microarrays are not

suitable [33]. Deep sequencing of metagenomes and metatranscriptomes broadens the landscape available for novel virus detection by many orders of magnitude. With the promise of third-gen sequencing [34], this uncharted landscape of sequence data will keep expanding, necessitating the need for novel tools for mining and annotating sequence data. We envision that as the accuracy of RNA sequencing technologies advance, the software that are used to assemble them will also upgrade and models such as ours that facilitate the annotation of the assembled data will be expected to catch up to these advancements.

## Methods

### Contriving features using k-mer abundance profiles

For the DNA letters 'A', 'C', 'G' and 'T', we use all possible k-mers of length 1 to 6 to derive feature scores for each sequence. We use DNA letters instead of RNA letters where Thymidine (T) is replaced by Uracil (U) because by convention most sequence data formats are encoded using DNA letters and this is also true for RNA transcripts and RNA genomes. The number of features for any sequence is thus the total number of possible k-mers, or in our case 5460, as shown below.

$$\sum_{i=1}^{6} 4^i = 5460$$

Through dynamic programming, our algorithm simulates the task of sequentially matching a k-mer against a sequence from start to end, going along the sequence like a sliding window, one ribonucleotide base at a time. For any sequence $q$ and k-mer $e$, we obtain a total count $C_j^q(e)$ when allowing exactly $j$ total mismatches or gaps.

Our final sequence score $W_q(e)$ for element $e$, and sequence $q$, is calculated by the formula given below where n is the length of sequence $q$.

$$W_q(e) = \sum_{j=0}^{3} \frac{C_j^q(e)}{2^j} / n$$

This final score is used as the pre-scaled value for feature $e$ in sequence $q$ in all constructed machine learning models in this study.

### Data preprocessing

We sourced all available virus sequences from NCBI [35]. Taxonomical data related to virus genomic architecture was retrieved from the ICTV website [36]. We found 3145 unique viruses having either ssRNA(+) and ssRNA(-) genomes, with 2862 belonging to the plus-strand group and 283 to the minus-strand. The complete human reference transcriptome was obtained from NCBI. All predicted and non-putative RNA sequences from the reference transcriptome were omitted. In compliance with the general length of animal viruses, to build a human/virus binary RNA classifier, only putative and expressed human RNA ranging from 2000 to 35000 in length were selected for the study. For sequences sharing greater than 85% similarity, only the longest sequence was selected for use while omitting all others. We were left with 4114 unique viral genomes and 15476 human RNA transcripts (S6 Table). The reverse complement of the RNA(+) strand in ssRNA(+) viruses is a mandatory intermediate in its replication and the reverse complement of the RNA(-) strand in ssRNA(-) viruses is a mandatory intermediate in both replication and the information pathway. We therefore included the reverse complements of the working RNA virus genomes in our data set.

## Libraries and parameters

Python's biopython module was used for all sequence manipulation tasks. For all statistical analysis we used the scipy.stats module in Python and scikit-learn was used for all machine learning implementations. Plots were drawn using matplotlib's pyplot. Train-test splits were always 4:1 for training set and test set respectively, unless specifically mentioned in the results. Standard scaling was employed to scale the features. 10-fold cross-validation was carried out within the training sets and both mean and standard deviation of the macro average f1 score and binary f1 score was reported. Similarly, for the test set, both macro averaged f1 scores and binary f1 scores were reported. For skewed classes it is classic convention to label the smaller class as the positive example. Hence, in all training and test examples in this study, viruses were set as the positive binary class. In the classification of positive and negative sense RNA viruses, there were 2862 examples in the positive RNA class and 283 examples in the negative RNA virus class. The negative sense RNA viruses were considered the positive binary class.

Logistic Regression models were trained with L2 penalty and 2000 max iterations. For the Support Vector Machine classifiers, both linear and RBF penalization were separately employed. The probability parameter for the SVC class of sklearn was set to True in order to acquire probability scores for drawing ROC curves where needed. The tree-based models, Decision Trees and Random Forest were trained using the default information entropy function as criterion. The number of estimators was set at 200 for Random Forest. For any parameters not explicitly stated, the library defaults were used.

## Testing and performance evaluation

For cross-validation and generalization tests 11351 putative, expressed mRNA transcripts between 2000 and 35000 sequence length and sharing no more than 85% similarity with any other transcript were selected from the *Mus musculus* whole transcriptome available in NCBI. As divergent viruses, 2532 viruses between the specified sequence length range not having positive or negative strand RNA genomes were selected. All information about the 77 virus families and mouse transcripts can be found in S3 Table. De novo RNA-Seq assembly data was sourced from experiments by Hölzer et al. [24] available at https://doi.org/10.17605/OSF.IO/5ZDX4.

The following single-number evaluation metrics were used as a measure of model performance.

**F1-score.** A measure of balance between precision and recall defined as $\frac{2 \times (precision \times recall)}{(precision + recall)}$.

**F1-binary.** The F1 score where only the predictions of the positive class are considered.

**F1-macro.** The macro-averaged F1 score where the unweighted mean for each class label is taken into account to adjust for an imbalance of class size.

**CV_bin and CV_macro.** Set of F1-binary and F1-macro scores for cross-validation tests respectively. Both the mean and standard deviation are reported.

**AUROC.** An abbreviation of Area Under the Receiver Operating Characteristic Curve. A graphical representation of the discriminatory capacity of a binary classifier. A random predictor would have an AUROC of 0.5.

**Accuracy.** Accuracy in this paper is specifically reserved for cases where the proportion of correct class predictions is reported. Hence, $accuracy = \frac{correct\ classifications}{total\ number\ of\ examples}$.

**Unary accuracy.** Accuracy for a data set where all relevant datapoints belong to the same class.

Command-line BLAST+ [25] was used for sequence alignments with blastn and blastp. All alignment tasks were carried out locally. For both blastn and blastp, false hits were first isolated

by query against a negative control database (i.e. human transcriptome for RNA viruses and our RNA virus sequence database for the de novo assembly data) and only informative hits were considered in the final counts. For blastp, cutoff for minimum ORF length was set at 100 bases. The command-line HMMER3 [37] application was employed for Hidden Markov Profile searches. The hmmsearch command was first carried out on virus protein family alignments known as vFams [22] against our negative control human transcriptome, to remove any alignment profiles that might be uninformative. Later, only the filtered informative protein profiles were used for the final search in metagenomic data. Informative vFams are deposited in the git repository as a *hmm* file.

The ten assembly tools that have been evaluated are BinPacker [38], Bridger [39], Shannon [40], rnaSPAdes [41], SPAdes [42], idba-tran [43], SOAPdenovo-Trans [44], Trans-Abyss [45], Trinity [46] and Oases [47]. All computational tasks were carried out on a third-generation Intel core-i5 machine having 8GB of RAM. All source-code, data and executed commands for this project including the trained classifiers are available on github at https://github.com/DeadlineWasYesterday/V-Classifier.

## Supporting information

**S1 Fig. Performance of the 18 classifiers when trained on different numbers of features.** (a) Logistic Regression Classifier on 3 Feature Sets. (b) Linear SVM Classifier on 3 Feature Sets. (c) RBF-Kernel SVM Classifier on 3 Feature Sets. (d) Decision Tree Classifier on 3 Feature Sets. (e) Random Forest Classifier on 3 Feature Sets. (f) Gaussian Naïve Bayes Classifier on 3 Feature Sets.
(JPG)

**S1 Table. All k-meric features ranked by feature importance instated by our feature selection flow-chart.**
(XLSX)

**S2 Table. Complete results for leave-one-family-out cross-validation procedure.**
(XLSX)

**S3 Table. Information of sequences used to test the model's ability to generalize.**
(CSV)

**S4 Table. False nucleotide blast hits across whole human transcriptome when queried against the RNA virus sequence database.**
(CSV)

**S5 Table. False nucleotide blast hits across whole mouse transcriptome when queried against the RNA virus sequence database.**
(CSV)

**S6 Table. Detailed information on training set sequences.**
(CSV)

## Acknowledgments

The findings and views expressed are that of the authors alone and none else.

## Author Contributions

**Conceptualization:** Md. Nafis Ul Alam, Umar Faruq Chowdhury.

**Data curation:** Md. Nafis Ul Alam.

**Formal analysis:** Md. Nafis Ul Alam.

**Investigation:** Md. Nafis Ul Alam.

**Methodology:** Md. Nafis Ul Alam.

**Project administration:** Md. Nafis Ul Alam.

**Resources:** Md. Nafis Ul Alam.

**Software:** Md. Nafis Ul Alam, Umar Faruq Chowdhury.

**Supervision:** Md. Nafis Ul Alam.

**Validation:** Md. Nafis Ul Alam.

**Visualization:** Md. Nafis Ul Alam, Umar Faruq Chowdhury.

**Writing – original draft:** Md. Nafis Ul Alam.

**Writing – review & editing:** Md. Nafis Ul Alam, Umar Faruq Chowdhury.

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
