## [Decision Letter · Decision Letter 0]

18 Aug 2020

PONE-D-20-22563

Short k-mer Abundance Profiles Yield Robust Machine Learning Features and Accurate Classifiers for RNA Viruses

PLOS ONE

Dear Dr. Alam,

Thank you for submitting your manuscript to PLOS ONE. After careful consideration, we feel that it has merit but does not fully meet PLOS ONE’s publication criteria as it currently stands. Therefore, we invite you to submit a revised version of the manuscript that addresses the points raised during the review process.

Authors need to prepare a new version of the manuscript in accordance with the recommendations and significant comments from the reviewers. Otherwise, the manuscript will not be published.

We look forward to receiving your revised manuscript.

Kind regards,

Ruslan Kalendar, PhD

Academic Editor

PLOS ONE

Journal Requirements:

Reviewers' comments:

Reviewer's Responses to Questions

**Comments to the Author**

1. Is the manuscript technically sound, and do the data support the conclusions?

Reviewer #1: Partly

Reviewer #2: No

2. Has the statistical analysis been performed appropriately and rigorously? 

Reviewer #1: N/A

Reviewer #2: Yes

3. Have the authors made all data underlying the findings in their manuscript fully available?

Reviewer #1: Yes

Reviewer #2: Yes

4. Is the manuscript presented in an intelligible fashion and written in standard English?

Reviewer #1: Yes

Reviewer #2: Yes

5. Review Comments to the Author

Reviewer #1: 

This manuscript addresses the problem of recognizing sequences of RNA viral origin within a data set of sequences. Their solution is a machine learning one, in which they derive from a sequence a set of features, each of which is a short $k$-mer, either its presence or its frequency, which is not clear. They use and evaluate several machine learning methods built using a Python machine learning package. The evaluation efforts are extensive and sufficient.

The authors make their Python software available through Github. Always capitalize Python.

There are two major problems with the exposition.

First, the authors never clearly define the classification problem they are trying to solve. After reading the entire manuscript, it is clearly a binary classification, but what exactly are the classes? Ask a computer scientist for help with properly defining a computational problem.

Second, the mathematical exposition on page 34 is the weakest part of the manuscript. The sentences, starting with the first two, make no mathematical sense. What do "mismatches and gaps" have to do with $k$-mers? Again, consult a computer scientist, preferably a theoretical computer scientist. Also, never use the blackboard abbreviation $\\forall$ in technical writing. That and similar notation is reserved for formal logic expressions. Write it out in English and do not put that in the displayed equation. The entire section needs a complete rewrite with more careful mathematical details.

Some minor issues follow.

Page 2. "De novo" is not hyphenated.

Page 3 and throughout. "numerous[1]" should be "numerous [1]". In general, there must be a space before a citation.

Page 3, line 48. "Often completely unbeknownst to us." is not a sentence. Make sure every sentence has a subject and a verb.

Page 3, line 49. "are astounding" should be "is astounding".

Page 3, line 57. "encoding;" should be "encoding,"

Page 4, lines 65-67. I verified this claim with a search of Web of Science.

Page 5, line 92. Do not use the possessive with inanimate things. So, "Scikit-learn's" should be rephrased "The scikit-learn". Make changes throughout.

Page 35, line 465. "dataset" should be "data set". Change throughout.

Page 37, lines 508-509. Capitalize Intel. "8GBs" should be "8GB".

Figure 1, "Layers of Feature Selection", suffers from poor resolution and poor readability. There is no need for the color. Just use black text in black boxes with white background for maximum readability.

Reviewer #2: 

Summary

This study applies a short k-mer sequence scores based machine-learning method for classifying RNA viruses from human transcripts. The short k-mer based features are selected and tested with six different classification models: Logistic Regression, SVM, Kernel SVM, Decision Tree, Random Forest and Gaussian Naïve Bayes Classifiers. The methods were mainly tested on (1) RNA viral genomes and human transcripts (2) de-novo assembled contigs from RNA-seq data, and achieved good AUC results on the first data set and reasonable results on the second one.

Major issues

1. The main issue is the experiment results lack comparisons. (1) All the results in this study are not compared with other machine learning based viral sequences classifiers, while there are a lot of those kinds of tools, especially those that are also based on k-mer features, like VirFinder. (2) For the results on RNA viral genomes and human transcripts (pager 7, line 113), while the tested models have good AUCs, the results are not compared with blast results or other machine-learning based methods, which does not prove the superiority of the proposed methods

2. For the models training and testing, while as stated a 25% split of training and testing data set is performed (page 7, line 118), no test metrics are reported. For Figure 3, why report the results of cross-validation but not the testing results?

3. The results on real RNA-seq assemblies seem not to be better than blast. On page 15, line 208, blast “yielded 135 false positives” on Human transcriptome assemblies. However, comparing to the results in Table 3, rbf-SVM68 has much more false positives comparing to blast (7067 even after length filtered)

4. How ssRNA(+) and ssRNA(-) are distinguished is not well explained. Usually contigs assembled from RNA-seq data contain both plus and minus strand sequences for a genome, can these contigs be successfully classified as ssRNA(+) or ssRNA(-)?

Minor issues:

1. In page 15, line 208, blast “yielded 135 false positives” on Human transcriptome assemblies, while in Supplementary Table 4, there are actually 243 sequences.

2. In page 8, line 130, AUROC is already used before the text “Area under the receiver operating characteristic curve”

3. In Figure 5, typo “Only humam_filtered”

4. Evaluation metrics need some explanation, like macro-averaged f1-score, binary f1-score, ROC AUC etc.

6. PLOS authors have the option to publish the peer review history of their article (what does this mean?). If published, this will include your full peer review and any attached files.

Reviewer #1: **Yes: **Lenwood S. Heath

Reviewer #2: No

---

## [Author Response · Author response to Decision Letter 0]

18 Aug 2020

Response to reviewer #1

Dear Dr. Heath,

Thank you for your valuable feedback. In this revision, we have tried our best to satisfy all previous shortcomings that had been pointed out.

Reviewer #1: The authors make their Python software available through Github. Always capitalize Python.

Response: All occurrences of the word Python were capitalized.

Reviewer #1: There are two major problems with the exposition. First, he authors never clearly define the classification problem they are trying to solve. After reading the entire manuscript, it is clearly a binary classification, but what exactly are the classes? Ask a computer scientist for help with properly defining a computational problem.

Response: We acknowledge this concern by adding the following changes in lines 73 and 79 of the introduction: 

“With our scoring function, we were able to calculate the profiles of short genetic elements (k-mers) for RNA virus genomes and human RNA transcripts. We trained a number of classical machine learning classifiers on our sequence-derived numerical data in an attempt to classify an assembled RNA sequence of unknown origin as either viral or cellular RNA.”

“Furthermore, we demonstrated the flexibility of this approach by attempting to classify an RNA virus genome sequence from NCBI as either a positive-strand (ssRNA(+) ) or a negative-strand (ssRNA(-) ) RNA virus.”

Reviewer #1: Second, the mathematical exposition on page 34 is the weakest part of the manuscript. The sentences, starting with the first two, make no mathematical sense. What do "mismatches and gaps" have to do with $k$-mers? Again, consult a computer scientist, preferably a theoretical computer scientist. Also, never use the blackboard abbreviation $\\forall$ in technical writing. That and similar notation is reserved for formal logic expressions. Write it out in English and do not put that in the displayed equation. The entire section needs a complete rewrite with more careful mathematical details.

Response: We have consulted with our colleagues that have a background in computer science and re-written the mentioned section hoping to make it more mathematically sound and comprehensible. The section is now as follows.

“

Contriving features using k-mer abundance profiles

For the DNA letters ‘A’, ’C’, ‘G’ and ‘T’, we use all possible k-mers of length 1 to 6 to derive feature scores for each sequence. We use DNA letters instead of RNA letters where Thymidine (T) is replaced by Uracil (U) because by convention most sequence data formats are encoded using DNA letters and this is also true for RNA transcripts and RNA genomes. The number of features for any sequence is thus the total number of possible k-mers, or in our case 5460, as shown below. 

∑_(i=1)^6▒4^i =5460

Through dynamic programming, our algorithm simulates the task of sequentially matching a k-mer against a sequence from start to end, going along the sequence like a sliding window, one ribonucleotide base at a time. For any sequence q and k-mer e, we obtain a total count C_j^q (e) when allowing exactly j total mismatches or gaps. 

Our final sequence score W_q (e) for element e, and sequence q, is calculated by the formula given below where n is the length of sequence q.

W_q (e)=(∑_(j=1)^3▒(¬C_j^q (e))/2^j )⁄n 

This final score is used as the pre-scaled value for feature e in sequence q in all constructed machine learning models in this study.

”

Reviewer #1: Page 2. "De novo" is not hyphenated.

Response: The hyphen was removed.

Reviewer #1: Page 3 and throughout. "numerous[1]" should be "numerous [1]". In general, there must be a space before a citation. 

Response: This (no space) is the default behavior of the citation manager. Nonetheless, the advised spaces were added.

Reviewer #1: Page 3, line 48. "Often completely unbeknownst to us." is not a sentence. Make sure every sentence has a subject and a verb.

Response: We have omitted the sentence.

Reviewer #1: Page 3, line 49. "are astounding" should be "is astounding".

Response: Corrected.

Reviewer #1: Page 3, line 57. "encoding;" should be "encoding,"

Response: Corrected.

Reviewer #1: Page 4, lines 65-67. I verified this claim with a search of Web of Science.

Response: We are yet to notice otherwise as well.

Reviewer #1: Page 5, line 92. Do not use the possessive with inanimate things. So, "Scikit-learn's" should be rephrased "The scikit-learn". Make changes throughout.

Response: Changes were made throughout the manuscript so as to not anthropomorphize the sklearn library.

Reviewer #1: Page 35, line 465. "dataset" should be "data set". Change throughout.

Response: We believe they can be used interchangeably. However, we have changed throughout as instructed.

Reviewer #1: Page 37, lines 508-509. Capitalize Intel. "8GBs" should be "8GB".

Response: Corrected.

Reviewer #1: Figure 1, "Layers of Feature Selection", suffers from poor resolution and poor readability. There is no need for the color. Just use black text in black boxes with white background for maximum readability.

Response: The figure has been modified to meet the mentioned criteria. We have maximized the contrast by using very light colors and black fonts instead of going completely black-and-white.

Response to reviewer #2

We appreciate the concerns and criticisms raised by reviewer #2. We have made some significant changes to address some issues, and will provide a very comprehensive clarification here on matters of rebuttal. 

Reviewer #2: 1. The main issue is the experiment results lack comparisons. (1) All the results in this study are not compared with other machine learning based viral sequences classifiers, while there are a lot of those kinds of tools, especially those that are also based on k-mer features, like VirFinder. (2) For the results on RNA viral genomes and human transcripts (pager 7, line 113), while the tested models have good AUCs, the results are not compared with blast results or other machine-learning based methods, which does not prove the superiority of the proposed methods

Response: The task of making comparisons to other published models had been on the back of our minds throughout the duration of the study. Towards the end, we felt that it became apparent that our models show superior performance. Because there were so few similar studies on the subject and because our manuscript was getting quite long for the topic, we decided not to include tabular or graphical comparisons.

We had gone over the VirFinder paper and it had also been cited in the manuscript. The reported performance of VirFinder varies significantly as a function of the lengths of k-mers used, the proportion of viral to bacterial contigs and the size of the contigs. We reported our performance metrics for 5460, 194 and 68 features respectively and got consistent results (Fig. 1 and Supplementary Fig. 1). We also tested the classifiers with varying proportions of sequence data (E.g. when cross-validating by leaving large families out, when testing on mouse transcripts and when testing our PN classifier on very small test sets). The AUROC for our models under varying parameters are notably better and more consistent than those reported in the VirFinder paper. However, our model performance does fluctuate similarly to VirFinder with varying sequence lengths (as evident from Fig. 5). 

Additionally, they (VirFinder) never test their model’s sequence discriminatory capacity on real data. They have used real data to try and predict healthy/disease state instead, which employs a very different workflow. The simulation dataset they test on, and report 0.90 to 0.97% AUROC curve is built on top of downloaded reference sequences, which is more analogous to our Mouse transcript and DNA virus test set, where we report 0.95 to 0.96% AUROC. It is important to note the following sentence from line 31 in the abstract: 

“For clean sequence data retrieved from curated databases, our models display near perfect accuracy, outperforming all similar attempts previously reported.”

Coming back to VirFinder- we can accept these metrics because our experiment is substantially different from the VirFinder study (they manually cleave the genomes to sample them, they classify bacteria against prokaryotic viruses, most prokaryotic viruses are DNA viruses[1], they use contigs whereas we use assemblies, they use reference-based tools whereas we use de novo assemblies, they use prediction categories, etc.) Our manuscript only claims to outperform ‘similar attempts from the past’. Programs like ViraMiner[2] and Bzhalava et al. 2018 [3] are more ‘similar’ in methodology as well as the goal to accomplish. 

In spite of this rebuttal, we have made significant and perhaps necessary changes to the manuscript. The following was added to the discussion section in line 324.

“

Machine learning attempts at sequence classification have been sparse in the past. But as of late, there is more activity in the field primarily due to advances brought about by next-gen sequencing and growing interest in metagenomics [21, 31, 32], symbionts [11] and microbial colonies [28]. Bzhalava et al. 2018 [21] attempted to detect viral sequences in human metagenomic data sets and although their classifier found very promising contrast on GenBank data, the model performed very poorly on a real metagenomic data set. Our models trained on NCBI data displayed 0.8 to 0.95 AUROC on real test data (Fig. 5), which is a big improvement form their AUROC of 0.51. k-mer based Random Forest models by the same team using k-mers of lengths 1 to 7 achieved the best AUROC of 0.875 with k-mers of length 6 [16], which is significantly lower than the metrics of our Random Forest models. VirFinder is a k-mer based machine learning tool that can identify prokaryotic viral contigs from metagenomic data [11]. The performance of VirFinder was evaluated with varying k-mer lengths, class skews and contig lengths. Our models trained on varying numbers of features and large class discrepancies display more consistent and robust performance metrics on average across all tests concerning NCBI data. The AUROC for our models on the divergent data set comprised of mouse transcripts and DNA viruses was 0.95 to 0.96 compared to their 0.90 to 0.98 on an analogous simulated data set. 

“

Reviewer #2: 2. For the models training and testing, while as stated a 25% split of training and testing data set is performed (page 7, line 118), no test metrics are reported. For Figure 3, why report the results of cross-validation but not the testing results?

Response: All results had been reported in the original manuscript. An unfortunate misunderstanding has occurred here. The f1bin and f1mac columns in Table 2 are the metrics for the test sets. These column names had also been explained on the footnotes of Table 2 in the original manuscript.

However, to avoid any misunderstanding by the audience, we have added the following section to our Methods section in line 523 to explain our single number evaluation metrics.

“

The following single-number evaluation metrics were used as a measure of model performance.

F1-score: A measure of balance between precision and recall defined as (2×(precision × recall))/((precision + recall)) .

F1-binary: The F1 score where only the predictions of the positive class are considered.

F1-macro: The macro-averaged F1 score where the unweighted mean for each class label is taken into account to adjust for an imbalance of class size.

CV_bin and CV_macro: Set of F1-binary and F1-macro scores for cross-validation tests respectively. Both the mean and standard deviation are reported.

AUROC: An abbreviation of Area Under the Receiver Operating Characteristic Curve. A graphical representation of the discriminatory capacity of a binary classifier. A random predictor would have an AUROC of 0.5.

Accuracy: Accuracy in this paper is specifically reserved for cases where the proportion of correct class predictions is reported. Hence, accuracy= (correct classifications)/(total number of examples) .

Unary accuracy: Accuracy for a data set where all relevant datapoints belong to the same class.

”

Additionally:

We have tried our best to be as transparent as possible with all our results. For example, we expected our model to perform better when we separately included the protein coding transcripts from viruses into the training data because coding regions are much more informative than other parts of the genome. However, it resulted in a decline in overall performance. This observation was mentioned in the discussion of the original transcript (now line 407) as follows: 

“Since all human training data was composed of gene transcripts, whereas the sequences in the virus training set were mostly comprised of non-genic regions, we hypothesized that this could be a cause for greater numbers of false negatives because the majority of virus sequences in the cell-virus incubate would also inevitably be viral gene transcripts as a consequence of the infectious activity of the Ebola virus, rather than the whole genome or sequencing with long non-genic regions. However, including RNA virus transcripts separately downloaded from NCBI into the training set only marginally improved performance on the Ebola virus and cell incubate sample (not discussed).”

Reviewer #2: 3. The results on real RNA-seq assemblies seem not to be better than blast. On page 15, line 208, blast “yielded 135 false positives” on Human transcriptome assemblies. However, comparing to the results in Table 3, rbf-SVM68 has much more false positives comparing to blast (7067 even after length filtered)

Response: The manuscript never claimed to have better performance than BLAST. We only mentioned that our classifiers outperform other similar attempts made in previous studies. In the concluding parts, our goal was to explain that even though BLAST is the gold standard for sequence classification and annotations, the tool is not resistant to false positives. Our classifier had much more false positives than BLAST, as apparent from our reported results. We explained in the results section that our classifier, in spite of having more false positives than BLAST, could correctly classify a large portion of the false positives that we acquire from BLAST search. The following excerpts are from lines 211, 216 and 220 of the results section. 

“In the Homo sapiens transcriptome assembly, carried out on GM12878 human gammaherpes virus 4 transformed B-lymphocyte cell lines, nucleotide blast query against our sequence database of ssRNA(+) and ssRNA(-) viruses yielded 135 false positives (Supplementary Table 4)…” 

“…Our 68 feature rbf kernel SVM model correctly classified 101 of these false positives giving us an accuracy of 75% on the unary data.” 

“Similarly, nucleotide blast had given 130 false positives on the Mus musculus transcriptome assemblies (Supplementary Table 5), whereas our model accurate classified 93 examples with 72% accuracy on the unary class.”

This observation has also been shared in the abstract in line 35:

“Our classifier was able to properly classify the majority of the false hits generated by BLAST and HMMER3 searches on the same data.”

This is what led to the idea behind Figure 10, that in order to acquire the best annotations, we must integrate the tools from different disciplines because any tool based on a single principle will have its shortcomings and yield some false positive results. This was explained in the conclusions of the original manuscript as follows (lines 432 and 439 in the revised manuscript). 

“Nucleotide blast is still the best tool at our disposal for characterization of unknown sequence data. But due to the homology omnipresent in biological sequences even between distinct orders of life, false positives and ambiguous results are always prevalent…” ”… Pertaining to the nonuniformity of the overlap between the results from our classifier and the tools it was compared to, we propose that a combinatorial approach employing multiple tools simultaneously is currently the best option for segregating sequence data by taxonomy (Fig 10).”

Reviewer #2: 4. How ssRNA(+) and ssRNA(-) are distinguished is not well explained. Usually contigs assembled from RNA-seq data contain both plus and minus strand sequences for a genome, can these contigs be successfully classified as ssRNA(+) or ssRNA(-)?

Response: Yes. For curated data (as is done in the manuscript), absolutely. The sole purpose of our PN classifier was to demonstrate the flexibility of our machine learning pipeline, which it competently does.

We have not done any classification tasks for classifying ssRNA(+) and ssRNA(-) sequences from the de novo assembly data. The PN classifier explained in the results sections was trained and tested entirely with well-annotated viral genome sequences that are deposited in NCBI. The information about the genome types of these viruses were tactfully sourced from the data available in the ICTV website[4] using computer algorithms in Python. This entire process is documented in the git repository. This was also mentioned in the methods section (now line 481 in the revised manuscript).

Reviewer #2: Minor issues: 1. In page 15, line 208, blast “yielded 135 false positives” on Human transcriptome assemblies, while in Supplementary Table 4, there are actually 243 sequences.

Response: The table has been fixed. In actuality, there were 135 false positives on the table. But the number of rows were 243 because of repeated entries. Simply dropping the duplicates gave us 135 data rows. Which we will now upload as an updated table.

Reviewer #2: 2. In page 8, line 130, AUROC is already used before the text “Area under the receiver operating characteristic curve”

Response: Changed to AUROC.

Reviewer #2: 3. In Figure 5, typo “Only humam_filtered”

Response: Fixed.

Reviewer #2: 4. Evaluation metrics need some explanation, like macro-averaged f1-score, binary f1-score, ROC AUC etc.

Response: Already addressed in the response to the second major point raised by reviewer #2. We have added a new section to the methods briefly describing the evaluation metrics.

1. McGrath, S. and D.v. Sinderen, Bacteriophage: genetics and molecular biology. 2007: Caister Academic Press.

2. Tampuu, A., et al., ViraMiner: Deep learning on raw DNA sequences for identifying viral genomes in human samples. PLOS ONE, 2019. 14(9): p. e0222271.

3. Bzhalava, Z., et al., Machine Learning for detection of viral sequences in human metagenomic datasets. BMC Bioinformatics, 2018. 19(1): p. 336.

4. Lefkowitz, E.J., et al., Virus taxonomy: the database of the International Committee on Taxonomy of Viruses (ICTV). Nucleic Acids Research, 2017. 46(D1): p. D708-D717.

---

## [Decision Letter · Decision Letter 1]

7 Sep 2020

Short k-mer Abundance Profiles Yield Robust Machine Learning Features and Accurate Classifiers for RNA Viruses

PONE-D-20-22563R1

Dear Dr. Alam,

We’re pleased to inform you that your manuscript has been judged scientifically suitable for publication and will be formally accepted for publication once it meets all outstanding technical requirements.

Kind regards,

Ruslan Kalendar, PhD

Academic Editor

PLOS ONE

---

## [Editor Report · Acceptance letter]

10 Sep 2020

PONE-D-20-22563R1 

Short k-mer Abundance Profiles Yield Robust Machine Learning Features and Accurate Classifiers for RNA Viruses 

Dear Dr. Alam:

I'm pleased to inform you that your manuscript has been deemed suitable for publication in PLOS ONE. Congratulations! Your manuscript is now with our production department. 

Kind regards, 

on behalf of

Dr. Ruslan Kalendar 

Academic Editor

PLOS ONE